# A trail of dark-matter-free galaxies from a bullet-dwarf collision

Pieter van Dokkum[1 ✉], Zili Shen[1], Michael A. Keim[1], Sebastian Trujillo-Gomez[2], Shany Danieli[3], Dhruba Dutta Chowdhury[1], Roberto Abraham[4], Charlie Conroy[5], J. M. Diederik Kruijssen[2], Daisuke Nagai[6] & Aaron Romanowsky[7,8]

The ultra-diffuse galaxies DF2 and DF4 in the NGC 1052 group share several unusual properties: they both have large sizes[1], rich populations of overluminous and large globular clusters[2–6], and very low velocity dispersions that indicate little or no dark matter[7–10]. It has been suggested that these galaxies were formed in the aftermath of high-velocity collisions of gas-rich galaxies[11–13], events that resemble the collision that created the bullet cluster[14] but on much smaller scales. The gas separates from the dark matter in the collision and subsequent star formation leads to the formation of one or more dark-matter-free galaxies[12]. Here we show that the present-day line-of-sight distances and radial velocities of DF2 and DF4 are consistent with their joint formation in the aftermath of a single bullet-dwarf collision, around eight billion years ago. Moreover, we find that DF2 and DF4 are part of an apparent linear substructure of seven to eleven large, low-luminosity objects. We propose that these all originated in the same event, forming a trail of dark-matter-free galaxies that is roughly more than two megaparsecs long and angled 7° ± 2° from the line of sight. We also tentatively identify the highly dark-matter-dominated remnants of the two progenitor galaxies that are expected[11] at the leading edges of the trail.

We begin with the assumption that it is not a coincidence that the ultra-diffuse galaxies DF2 and DF4 in the NGC 1052 group have the same set of otherwise-unique properties and that they were in close proximity to one another at the time of their formation. With that assumption, collisional formation is implied by their present-day radial velocities and three-dimensional locations. The geometry is shown in Fig. 1a. The relative radial velocity of the galaxies is high, 358 km s$^{-1}$, which is three times the velocity dispersion of the NGC 1052 group (about 115 km s$^{-1}$). Furthermore, although the two galaxies are separated by only 0.24 Mpc in the plane of the sky[3], a differential tip of the red giant branch (TRGB) analysis has shown that they are 2.1 ± 0.5 Mpc apart along the line of sight[15,16], which is five times the virial radius of NGC 1052 (about 400 kpc; refs. [16,17]). In the context of a shared origin of both galaxies, their radial velocities are consistent with their line-of-sight distances, that is, the closest galaxy (DF4) is moving towards us (with respect to the mean velocity) and the farthest galaxy (DF2) is moving away from us. Tracing their line-of-sight positions back in time, we infer that they must have formed in a high-velocity encounter. The minimum time since that encounter is about 6 Gyr, for constant motion along the line of sight.

The situation at that point, a ≳300 km s$^{-1}$ encounter between two galaxies in the gas-rich environment of a young group, is a close match to the initial conditions of mini-bullet-cluster[14] scenarios that have previously been proposed for the formation of DF2 and DF4[11–13]. In a near head-on collision between two gas-rich galaxies, the collisional gas can be shocked and separated from the collisionless dark matter and

pre-existing stars[11,12]. The accompanying star formation favours massive clumps in highly compressed gas[11,13], producing the unusual globular clusters and the lack of dark matter. Galaxies that form this way are initially compact[13], in apparent conflict with the large half-light radii of DF2 and DF4. However, intense feedback accompanying the formation of the globular clusters is expected to increase the sizes of the newly formed galaxies, an effect that is particularly efficient when the stars are not bound by a dark-matter halo[18,19]. These previous studies focused on the formation of a single new galaxy at the collision site, but we propose that both DF2 and DF4 were formed in a single 'bullet dwarf' event. An illustration of the proposed scenario is shown in Methods.

We have sufficient information to construct a plausible model for the geometry and timing of the collision. It is likely that the event took place near the central elliptical galaxy, NGC 1052, as it is roughly halfway between DF2 and DF4 in projection and its deep potential well is conducive to high-speed interactions. It is also likely that at least one of the progenitor galaxies was a satellite of NGC 1052 as the probability of a collision of two unbound galaxies is extremely small[12]. We assume that the second progenitor was not bound to NGC 1052, and either on first infall or a satellite of another massive galaxy in the group[20]. DF4 has a velocity difference of −43 km s$^{-1}$ with respect to NGC 1052, whereas the velocity of DF2 is +315 km s$^{-1}$. In Fig. 1b, we show a configuration for DF2, DF4 and NGC 1052 that satisfies these constraints. Progenitor 1 arrived in the vicinity of NGC 1052 with a high (>300 km s$^{-1}$) relative velocity and collided with progenitor 2, which was on a bound orbit.

[1]Department of Astronomy, Yale University, New Haven, CT, USA. [2]Astronomisches Rechen-Institut, Zentrum für Astronomie der Universität Heidelberg, Heidelberg, Germany. [3]Department of Astrophysical Sciences, Princeton University, Princeton, NJ, USA. [4]Department of Astronomy and Astrophysics, University of Toronto, Toronto, Ontario, Canada. [5]Harvard-Smithsonian Center for Astrophysics, Cambridge, MA, USA. [6]Department of Physics, Yale University, New Haven, CT, USA. [7]Department of Physics and Astronomy, San José State University, San Jose, CA, USA. [8]University of California Observatories, Santa Cruz, CA, USA. ✉e-mail: pieter.vandokkum@yale.edu

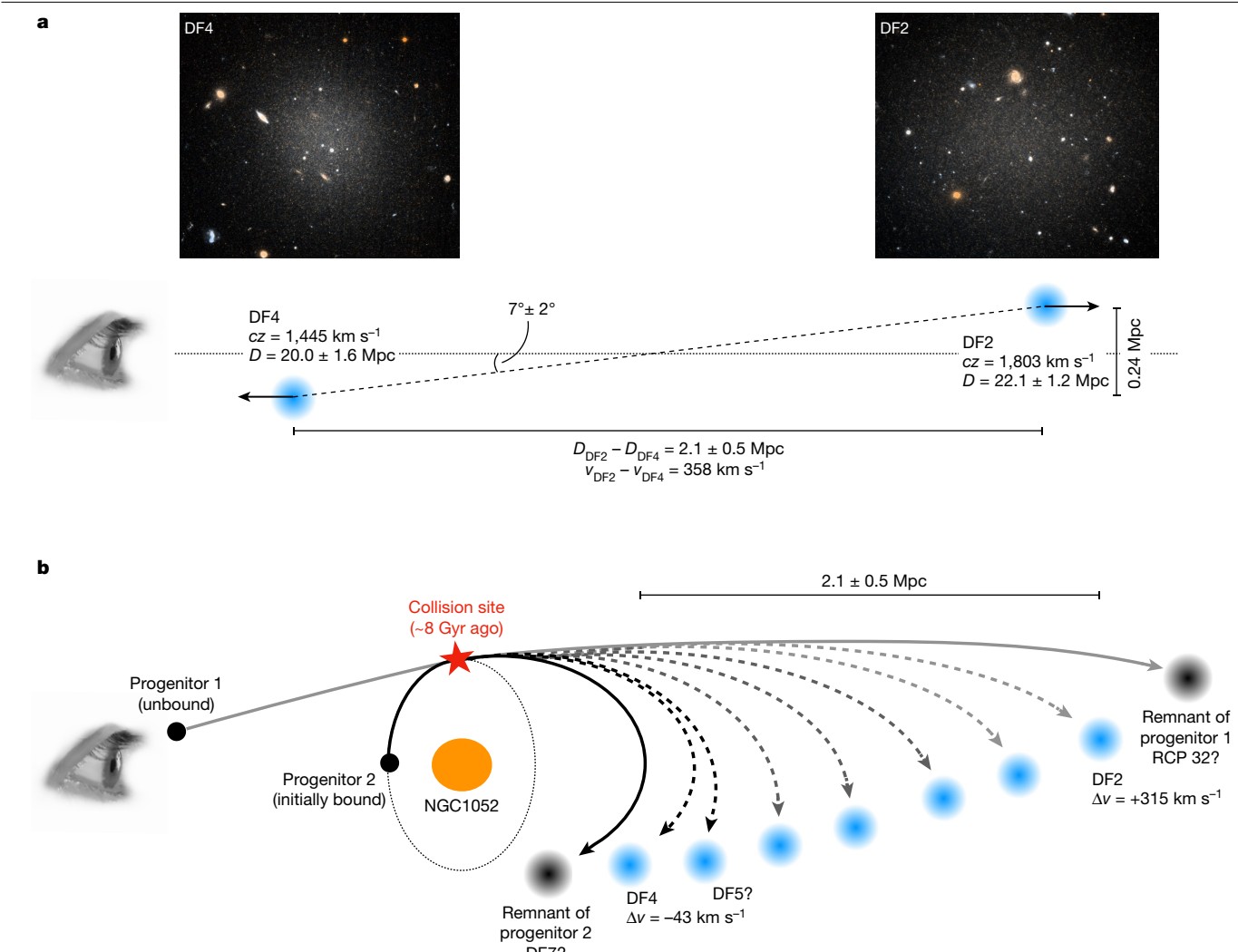

**Fig. 1 | Geometry of the DF2 and DF4 system. a**, The radial velocity difference between DF2 and DF4 is 358 km s⁻¹ and this large velocity difference is accompanied by a large line-of-sight separation of 2.1 ± 0.5 Mpc (refs. [15,16]). The geometry implies that the galaxies are moving away from each other. Tracing their positions back in time, we infer that they were formed in a high-speed encounter ≥6 Gyr ago. **b**, Example of a collisional scenario involving NGC 1052. Velocities are given with respect to that galaxy ($cz$ = 1,488 km s⁻¹). An infalling gas-rich galaxy on an unbound orbit collided with a satellite of NGC 1052 about 8 Gyr ago, leading to two dark remnants (possibly RCP 32 and DF7), DF2 and DF4, and three to seven other dark-matter-free galaxies.

The collision produced DF2 and DF4, with DF2's velocity and orbit similar to progenitor 1 and DF4's similar to progenitor 2. DF2 is currently unbound and about 3 Mpc behind the group, whereas DF4 remained bound and has begun falling back. This geometry is not unique but it is similar to examples that have been explored in simulations[12]. It is also consistent with the near-identical tidal distortions of the two galaxies[21]: in the geometry of Fig. 1b, DF2 and DF4 were at the same distance from NGC 1052 when they were formed, and as neither galaxy experienced a stronger tidal field afterwards, their morphologies have remained the same. In this model, DF2 has travelled about 3 Mpc since the collision with an average velocity with respect to NGC 1052 that is probably higher than its present-day value of 315 km s⁻¹. Assuming an average post-collision velocity of $\langle v \rangle \approx 350$ km s⁻¹ dates the collision to about 8 Gyr ago, in excellent agreement with the ages of the globular clusters and the diffuse light in DF2 (9 ± 2 Gyr, as measured from optical spectra[2,22]).

We further investigate the possible joint formation of DF2 and DF4 by examining the spatial distribution of galaxies along the DF2–DF4 axis. Given the complex gas distribution during and after the event, it may be that more than two dark-matter-deficient objects were formed in the wake of the collision[12]. Furthermore, the bullet-dwarf scenario predicts the existence of two dark-matter-dominated objects that are the remnants of the progenitor galaxies. These should precede DF2, DF4 and any other dark-matter-deficient galaxies along their path, as they have the highest velocities relative to the barycentre[11].

The spatial distribution of galaxies with magnitude $g > 16.5$ in a recently compiled catalogue of the NGC 1052 field[23] is shown in Fig. 2a. An objective search for linear features in this distribution is performed using a discrete implementation of the Hough transform[24]. The transform is shown in Fig. 2b. There is a clear peak with 11 galaxies on a line, corresponding to the relation

$$\triangle \text{Dec.} = 0.45 \triangle \text{RA} - 2.8, \tag{1}$$

in units of arcminutes north and west of NGC 1052 (where Dec. is declination and RA is right ascension). This relation is shown by the line in Fig. 2c. Both DF2 and DF4 are in the sample of 11 galaxies. The probability that the peak arose by chance is 3%, and the probability that the peak arose by chance and that both DF2 and DF4 are part of it is 0.6% (Methods). Before turning to the properties of the galaxies in the trail, we note that the Hough transform provides post hoc validation of our initial assumption that DF2 and DF4 are related to each other.

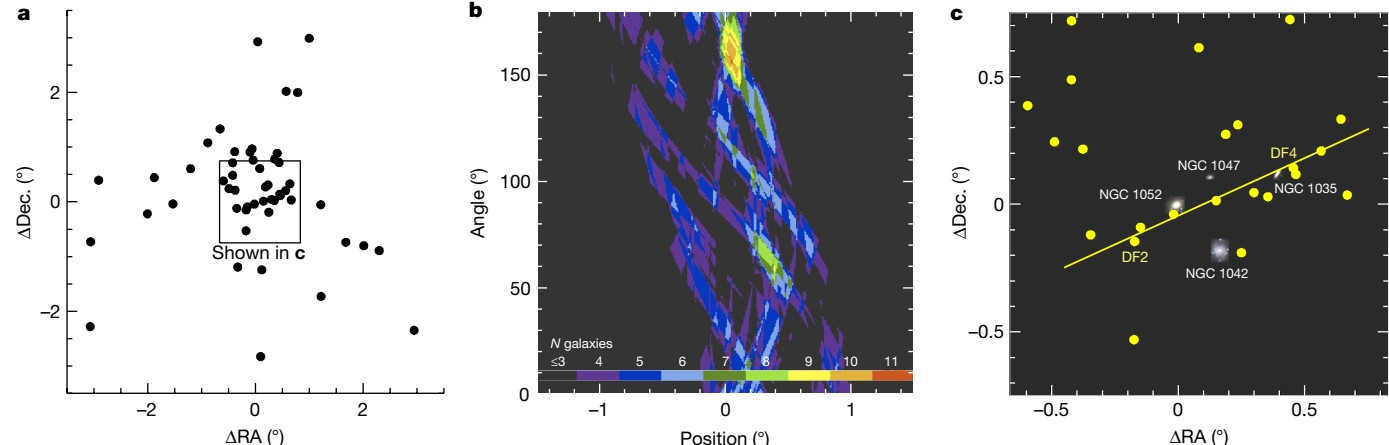

**Fig. 2 | A linear feature in the spatial distribution of faint galaxies in the NGC 1052 field. a**, Distribution of galaxies with $g > 16.5$ (black circles) from a recent compilation of low-surface-brightness objects in the NGC 1052 field[23]. The positions are with respect to the coordinates of NGC 1052, the central bright elliptical galaxy in the group. **b**, Hough transform[24] of the spatial distribution. The peak corresponds to a line that has 11 galaxies located within $\pm 30$ kpc. The significance of the feature, as determined from randomized realizations of the data, is 97%. **c**, Zoom of **a**, shown with the four brightest members of the NGC 1052 group. Yellow circles correspond to the positions of the galaxies in the box shown in **a**. The orientation and offset of the line corresponds to the location of the peak of the Hough transform. Both DF2 and DF4 are part of the linear feature.

Images of the 11 galaxies that are part of the trail are shown in Fig. 3. The average galaxy density in the central projected radius from NGC 1052 $R < 30'$ implies that $2 \pm 2$ of the 11 galaxies are chance projections, and we infer that there are 7–11 galaxies in the structure. Besides DF2 and DF4, other galaxies in the trail are also unusually large for their luminosity. The relation between size and apparent magnitude for faint galaxies in the NGC 1052 group is shown in Fig. 4a. After subtracting a simple linear-least-squares fit to the running median (dashed line), we find that galaxies in the trail are on average 26% larger than the rest of the sample. The Wilcoxon probability that the trail galaxies and the rest of the sample are drawn from the same size distribution is <1%. The spatial distribution of galaxies colour-coded by their (magnitude

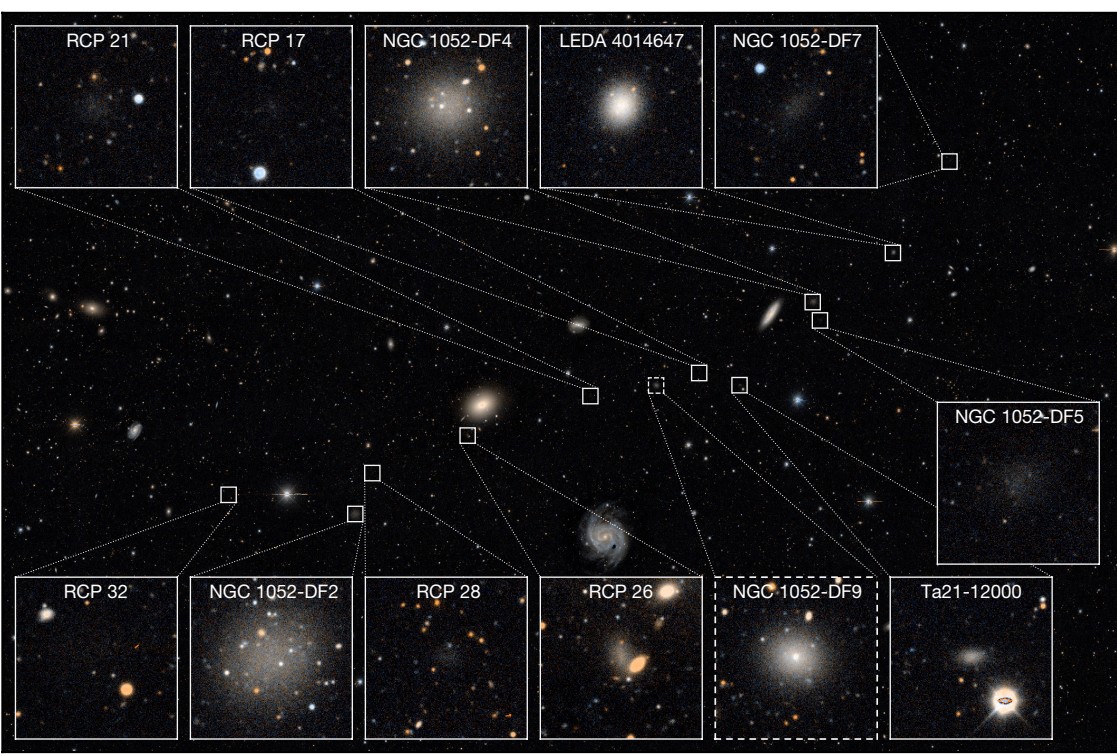

**Fig. 3 | Galaxies on the DF2–DF4 axis.** Legacy survey image of the central region of the NGC 1052 group, highlighting the 11 galaxies that are part of the trail according to the Hough transform. Several ($2 \pm 2$) of these are expected to be chance projections. LEDA 4014647 is a candidate for an interloper (that is, an unrelated group member) given its brightness and relative compactness. Its radial velocity was listed as $1,680 \pm 60$ in earlier SDSS releases (Data Release 3) but was later erroneously revised to a $z = 0.7$ quasi-stellar object (Data Release 16).

Judging from morphology alone, RCP 26 and Ta21-12000 may also be chance projections. Besides DF2 and DF4, RCP 32, DF5 and DF7 all satisfy or nearly satisfy the ultra-diffuse galaxy criteria. RCP 32 and DF7 are candidates for the two dark-matter-dominated remnants that have been predicted to precede dark-matter-deficient galaxies along the post-collision trajectory[11]. We also highlight DF9 (SDSS J024007.01−081344.4), a galaxy with a bright star cluster that falls on the trail but is not part of the objectively selected sample[23].

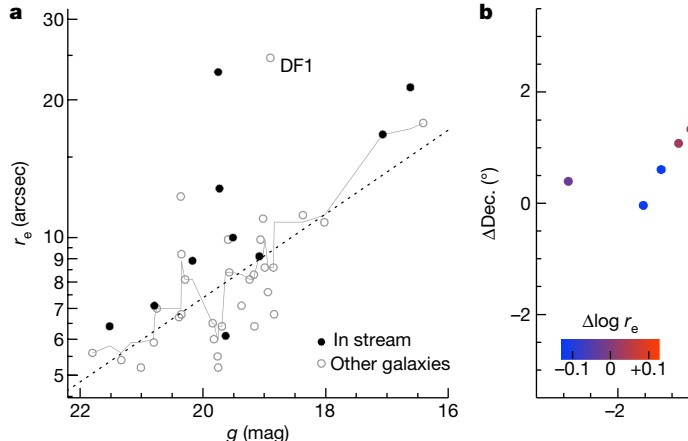

**Fig. 4 | The sizes of galaxies on the DF2–DF4 axis. a**, The apparent size–apparent magnitude relation for low-surface-brightness objects in NGC 1052, using recent measurements from Legacy data[23]. Galaxies that are part of the trail are labelled with solid symbols. The thin solid line is a running median, with $N = 7$. The dashed line is a least-squares fit to the running median and has

the form $\log r_e = -0.09(g - 20) + 0.87$ kpc. Galaxies in the trail are typically larger than other galaxies of the same magnitude. **b**, Distribution of galaxies colour-coded by their location with respect to the least-squares fit. The largest galaxies in the group are preferentially located along the DF2–DF4 axis.

dependent) relative size is shown in Fig. 4b. The unusual prevalence of large, low-surface-brightness galaxies in the central regions of the NGC 1052 group has been noted previously[1,23]; here we propose that the bullet-dwarf event was responsible for it.

We highlight three large and unusual galaxies that are part of the sample of 11. DF5 looks very similar to DF2 and DF4 but has a much lower surface brightness[1,21]. It is so close in projection to DF4 (only 1.6′ away) that both galaxies were observed in the same Hubble Space Telescope (HST) Advanced Camera for Surveys (ACS) pointing[1,9]. In the bullet-dwarf scenario, this puzzling object is readily explained as another dark-matter-free fragment that formed in the aftermath of the collision. RCP 32 and DF7 are the farthest away from the centre of the structure, and are 'ahead' of DF2 and DF4, respectively. RCP 32 is an extremely faint ultra-diffuse galaxy and may have a globular cluster system[23]. DF7 is an elongated ultra-diffuse galaxy that was previously observed with the HST[1] (Methods). We tentatively identify RCP 32 and DF7 as the candidate remnants of the original, pre-collision galaxies. In the geometry of Fig. 1b, RCP 32 could be the remnant of progenitor 1, the gas-rich object that was not bound to NGC 1052, and the brighter galaxy DF7 could be the remnant of progenitor 2, the galaxy that was a satellite of NGC 1052 at the time of collision. In this context, the elongation can be understood as the effect of the strong tidal forces at the time of collision[25].

The scenario that is proposed here makes predictions for the properties of the collision products, and further observations can test and refine this explanation. As the formation of DF2 and DF4 was triggered by a single event, the ages of the globular clusters of DF4, which have not yet been measured, should be identical to those of the clusters in DF2 ($9 \pm 2$ Gyr; refs. [2,22]). A stringent and model-independent test is to directly compare the (averaged) spectra of clusters in the two galaxies; any clear differences could falsify our model, particularly differences in age-sensitive spectral features. Turning to other galaxies on the trail, their kinematics are predicted to be consistent with baryon-only models—except RCP 32 and DF7, which could show evidence for an unusually low baryon fraction. Furthermore, the radial velocities of trail galaxies should follow the approximate relation $cz \approx 1{,}700 - 10\Delta\mathrm{RA}$ km s$^{-1}$, where $\Delta$RA is defined as in equation (1), $c$ is the speed of light and $z$ is the redshift. Similarly, their line-of-sight distances are predicted to be $D \approx 21.5 - 0.06\Delta\mathrm{RA}$ Mpc, for TRGB distances that are calibrated to $D = 22.1$ Mpc for DF2[16]. We note that these relations probably have considerable scatter and are not expected to be linear for most geometries. Also, 0–4 galaxies of the 11 are expected to be interlopers (Fig. 3), and

some galaxies that seem to be off the trail may in fact be part of it due to the foreshortening (Fig. 1a). This may be the case for DF1, a very large and diffuse galaxy (marked in Fig. 4) that is only 14′ off the DF2–DF4 axis and is elongated towards the centre of the trail.

Bullet-dwarf collisions hold the potential to constrain the self-interaction cross-section of dark matter. Modelling of the bullet cluster has provided an upper limit[26], but as self-interacting dark matter was introduced to explain the 'cored' dark-matter density profiles of low-mass galaxies[27], it is important to measure the cross-section on small scales[28]. Quantitative constraints will probably require more than a single example of a bullet dwarf. Encouragingly, these events are probably more common than the collision that produced the bullet cluster[29]; a search for plausible DF2 and DF4 progenitors in the IllustrisTNG cosmological simulation[30] produced 248 head-on high velocity collisions in a 100³-Mpc³ volume[12], corresponding to about 8 within $D < 20$ Mpc.

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

# Article

## Methods

### Illustration of the collision scenario

The proposed scenario for the formation of DF2, DF4 and the other trail galaxies is shown in Extended Data Fig. 1. As discussed in the main text, the scenario is a combination of the original idea that a bullet-dwarf collision might have formed DF2 and/or DF4[11]; the results from subsequent hydrodynamical simulations, showing that multiple dark-matter-free clumps can form in such a collision[12] and that the formation of massive star clusters is indeed promoted[13]; and the independent finding that feedback from massive cluster formation in these conditions leads to a rapid expansion of the galaxies[18].

### Faint galaxy sample

We make use of a recently compiled catalogue of low-surface-brightness objects in the NGC 1052 field[23], augmented by a catalogue of all brighter galaxies with redshifts in the range of $1,000$ km s$^{-1}$ $< cz < 2,000$ km s$^{-1}$ that is provided in the same study. Reference [23] makes use of the publicly available Dark Energy Camera Legacy Survey (DECaLS) dataset[31]. The galaxies were initially identified with a combination of automated techniques and visual inspection, with the majority coming from visual inspection. Their structural parameters were measured with IMFIT[32]. We caution that the DECaLS dataset suffers from sky subtraction errors around low-surface-brightness galaxies, and that this may bias the size measurements. The main point of Fig. 4 is a relative comparison of the sizes of galaxies on and off the trail and this should be more robust than the absolute size measurements.

### Velocity dispersion of the NGC 1052 group

We use the latest compilation of radial velocities in the NGC 1052 field[23] for an updated value of the velocity dispersion of the group. Table 2 of ref. [23] contains 30 galaxies with redshifts $cz < 2,000$ km s$^{-1}$. Two were removed: DF2, as it is almost certainly not bound to the group, and LEDA 4014647. LEDA 4014647 was assigned a radial velocity of $1,680 \pm 60$ km s$^{-1}$ in earlier Sloan Digital Sky Survey (SDSS) releases (Data Release 3), but its redshift was later revised to $z = 0.7$ (Data Release 16). A visual inspection of the SDSS spectrum shows no clear features. Using the biweight estimator[33], we find a central velocity for the remaining 28 galaxies of $\langle cz \rangle = 1,435 \pm 20$ km s$^{-1}$ and a line-of-sight velocity dispersion of $\sigma = 115 \pm 15$ km s$^{-1}$.

### The Hough transform

We use the Hough transform to look for linear features in the galaxy distribution, a standard method for detecting lines in images[24]. The transform provides the number of galaxies along all possible directions, characterized by an angle and a distance from the centre. A width and maximum linear extent have to be chosen; we use $\pm 30$ kpc ($\pm 5.2'$) for the width and $<400$ kpc ($69'$) for the linear extent. Although the exact number of galaxies that the Hough transform associates with the linear feature depends on the precise limits that are chosen, the qualitative results are not sensitive to them. In Fig. 2b, the orientation of the line is on the vertical axis and offset with respect to NGC 1052 on the horizontal axis.

### Statistical significance of the trail

We use simulations to assess the probability that the alignment of the 11 galaxies arose by chance. We generate $N = 1,000$ realizations of the $(x, y)$ pairs by maintaining the angular distance from NGC 1052 for each pair and randomizing the angle. This procedure ensures that the density profile of the sample is maintained for all realizations. We then create Hough transforms for all realizations and determine how often the strongest linear feature contains $\geq 11$ galaxies. We find that the probability of a chance alignment of $\geq 11$ galaxies is 3%.

This calculation assumes that galaxies are oriented randomly with respect to NGC 1052, and does not take into account anisotropy associated with the filamentary structure of the cosmic web[34,35]. Galaxy groups are generally not spherical but have an average projected axis ratio of 0.77 (ref. [36]). We examined the large-scale structure in the NGC 1052 field using a recently compiled catalogue of galaxies[23] in this general area. Selecting all low-surface-brightness galaxies that were identified in that study plus all bright galaxies with $cz < 2,000$ km s$^{-1}$ gives a sample of 72 probable group members. Their distribution is shown in Extended Data Fig. 2. The smooth density field was calculated with the non-parametric kernel density estimator[37]. There is no evidence for large-scale structure associated with the trail. In fact, there are no galaxies in the trail direction in the outskirts of the group, and the overall orientation of the group is perpendicular to the trail. The assumption of isotropy is therefore slightly conservative, in the sense that more galaxies will be scattered towards the line than away from it.

Finally, we note that the probability that there is a chance alignment and that it is a coincidence that both DF2 and DF4 are part of it is very low. This joint probability can be calculated directly for the isotropic case: of the 31 simulations that have $\geq 11$ aligned galaxies only 6 have both DF2 and DF4 in the sample, corresponding to a combined probability of the observed arrangement of 0.6%.

### A 12th low-surface-brightness dwarf galaxy on the trail

Visual inspection of the DECaLS imaging[31] readily shows that there is a fairly prominent 12th galaxy that is part of the apparent trail. The object is SDSS J024007.01−081344.4 (ref. [31]); it was previously pointed out as a likely low-luminosity group member with a central star cluster[38]. It is not in the objective catalogue that we use for the main analysis[23]. This may be because of its redshift in the SDSS database (it is erroneously listed as a $z = 0.933$ active galactic nucleus) or because the light from the central cluster moved the object outside of the size and surface-brightness criteria. We refer to the galaxy as DF9 as that was the catalogue number in our initial Dragonfly catalogue[1]. We do not use the galaxy in the objective analysis but we show its DECaLS image in Fig. 3. For convenience, we provide the coordinates of all trail galaxies in Extended Data Table 1.

### HST imaging of the candidate dark galaxy DF7

DF7 is at one of the leading edges of the trail, 'ahead' of DF4. The galaxy was observed with HST/ACS as part of an exploratory survey of Dragonfly-identified low-surface-brightness galaxies in several groups[1]. The observations constituted two orbits, one orbit in F606W and one orbit in F814W. In Extended Data Fig. 3, we show the HST imaging at two different contrast levels. The galaxy is elongated and appears distorted, with the elongation in the direction of DF4. DF7's apparent distortion, combined with its location at the leading edge of the trail, lead us to speculate that the galaxy is the highly dark-matter-dominated remnant of one of the two progenitor galaxies. We note that DF7 may be largely disrupted in this interpretation: the observed[1] axis ratio is $b/a = 0.42$, but given the extreme foreshortening of the geometry the intrinsic axis ratio could be a stream-like approximately 1:20.

### Other proposed scenarios

The joint formation of DF2 and DF4 in a bullet-dwarf event explains their lack of dark matter, large sizes, luminous and large globular clusters, striking similarity, large distance between them, large radial velocity difference, and the presence of a trail of other galaxies on the DF2–DF4 axis. Here we briefly discuss other scenarios that have been proposed to explain the properties of DF2 and DF4.

Initially, follow-up studies focused on possible errors in the measurements, either in the masses[39] or in the distances of the galaxies[40,41]. However, with four independent velocity dispersion measurements[3,8–10] (three for DF2 and one for DF4) and TRGB distances from extremely deep HST data[15,16], these issues have now largely been settled.

Most astrophysical explanations centre on the absence of dark matter only, and invoke some form of extreme tidal interaction (with NGC 1052 or other galaxies) to strip the dark matter (along with a large fraction of

the initial stellar population)[42–45]. These models do not explain the low metallicity of the galaxies, why there are two nearly identical objects in the same group, the newly discovered trail, or their overluminous and too-large globular clusters. The globular clusters, which have the same age (within the errors) as the diffuse light[22], show that the galaxies were formed in an unusual way and did not merely evolve in an unusual way. Besides the bullet scenario, the only model that explains the globular clusters is a study of star formation in galaxies that are in the tails of the scatter in the halo mass–stellar mass relation[18,19]. This model has ad hoc initial conditions and does not account for the presence of two near-identical galaxies, but the key aspects of it (the formation of luminous globular clusters in a compact configuration and the subsequent puffing up of the galaxies owing to feedback) probably apply to the collision products in the bullet scenario (see main text).

It has recently been suggested that DF2 and DF4 are entirely unrelated, with DF4 being stripped of its dark matter by NGC 1035, which is near it in projection, and DF2 a face-on disk galaxy with a normal dark-matter content[46,47]. The association of DF4 with NGC 1035 is not seen in all datasets[21], and there is no compelling evidence that DF2 is a disk[7]. Furthermore, the globular clusters and the trail remain unexplained, and there is the question of the likelihood that DF2 and DF4 have entirely different explanations but coincidentally share several otherwise-unique properties.

## Data availability

The HST data for DF7 are available in the Mikulski Archive for Space Telescopes (MAST; http://archive.stci.edu), under programme ID 14644. The Legacy Survey data shown in Fig. 3 are available at https://www.legacysurvey.org/. All other data that support the findings of this study are available in published studies that are referenced in the text.

## Code availability

We have made use of standard data analysis tools in the Python environment.

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

**Acknowledgements** A.R. was supported by National Science Foundation grant AST-1616710 and as a Research Corporation for Science Advancement Cottrell Scholar; J.M.D.K. acknowledges funding from the German Research Foundation (DFG) in the form of an Emmy Noether Research Group grant no. KR4801/1-1. J.M.D.K.; and S.T.-G. acknowledge funding from the European Research Council (ERC) under the European Union's Horizon 2020 research and innovation programme via the ERC Starting Grant MUSTANG (grant agreement number 714907). Support from Space Telescope Science Institute grants HST-GO-14644, HST-GO-15695 and HST-GO-15851 is acknowledged. C. van Dokkum created Extended Data Fig. 1 and the eye photograph in Fig. 1.

**Author contributions** P.v.D. led the analysis and wrote the manuscript. Z.S. performed the relative distance measurement (published in ref. [16]) that is at the basis of the study. M.A.K. created Fig. 3 and measured the structural parameters for DF5. S.T.-G. created an early version of Fig. 1b. All authors commented on the manuscript and aided in the interpretation.

**Competing interests** The authors declare no competing interests.

**Additional information**
**Correspondence and requests for materials** should be addressed to Pieter van Dokkum.

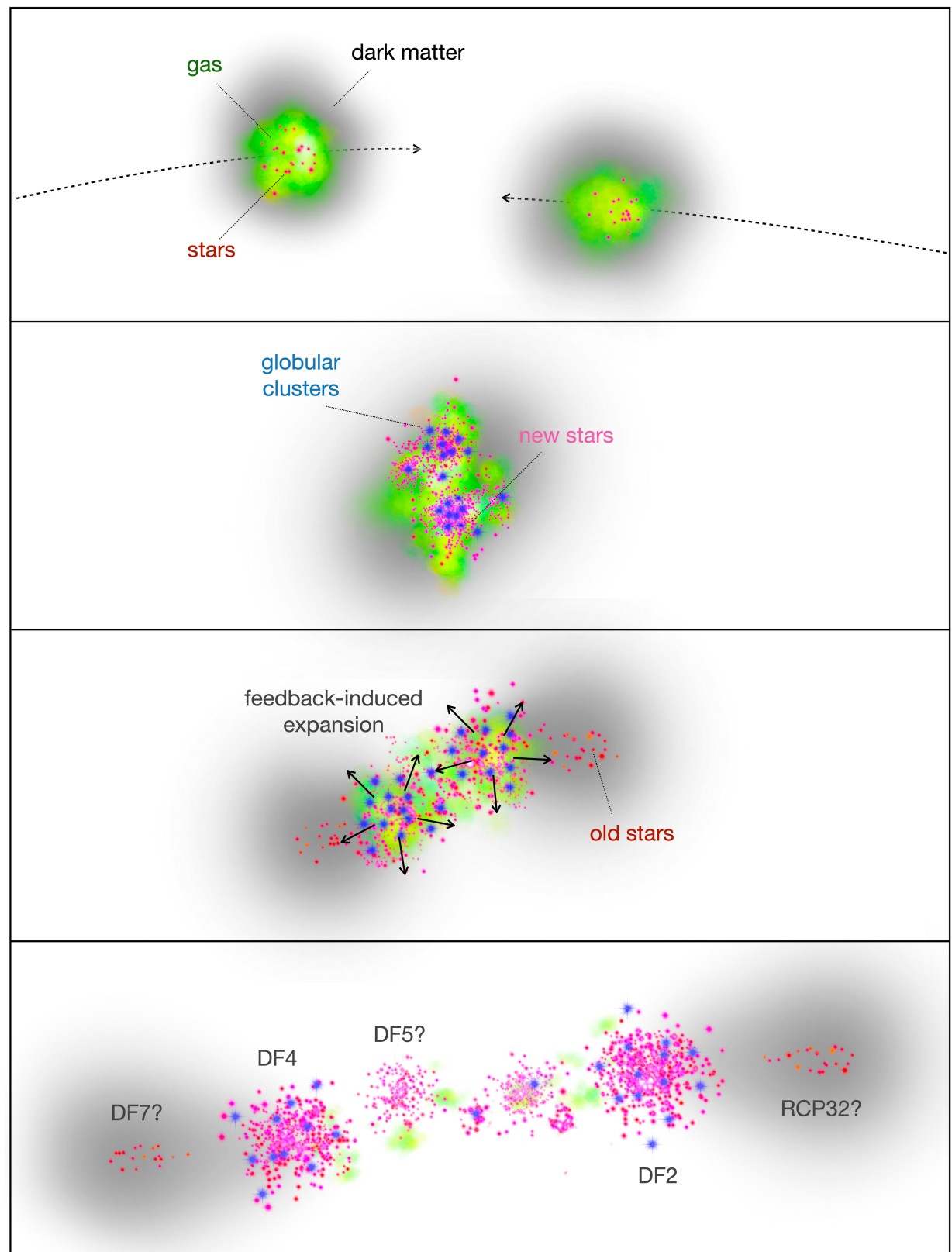

**Extended Data Fig. 1 | Illustration of the proposed formation scenario of DF2 and DF4.** Two gas-rich dwarf galaxies experience a high-speed encounter with a small impact parameter (top). Following previous studies[11–13] the collisional gas gets stripped and shocked at closest approach, and forms stars at a prodigious rate with a bias towards massive clumps (second from top). The dark matter and previously formed stars are tidally distorted but continue ahead of the newly forming galaxies (third from top). Feedback in the absence of a dark-matter halo leads to expansion of the newly formed galaxies[18]. Most of the newly formed stellar mass is in two clumps, but several lower mass galaxies have also formed in the wake (bottom).

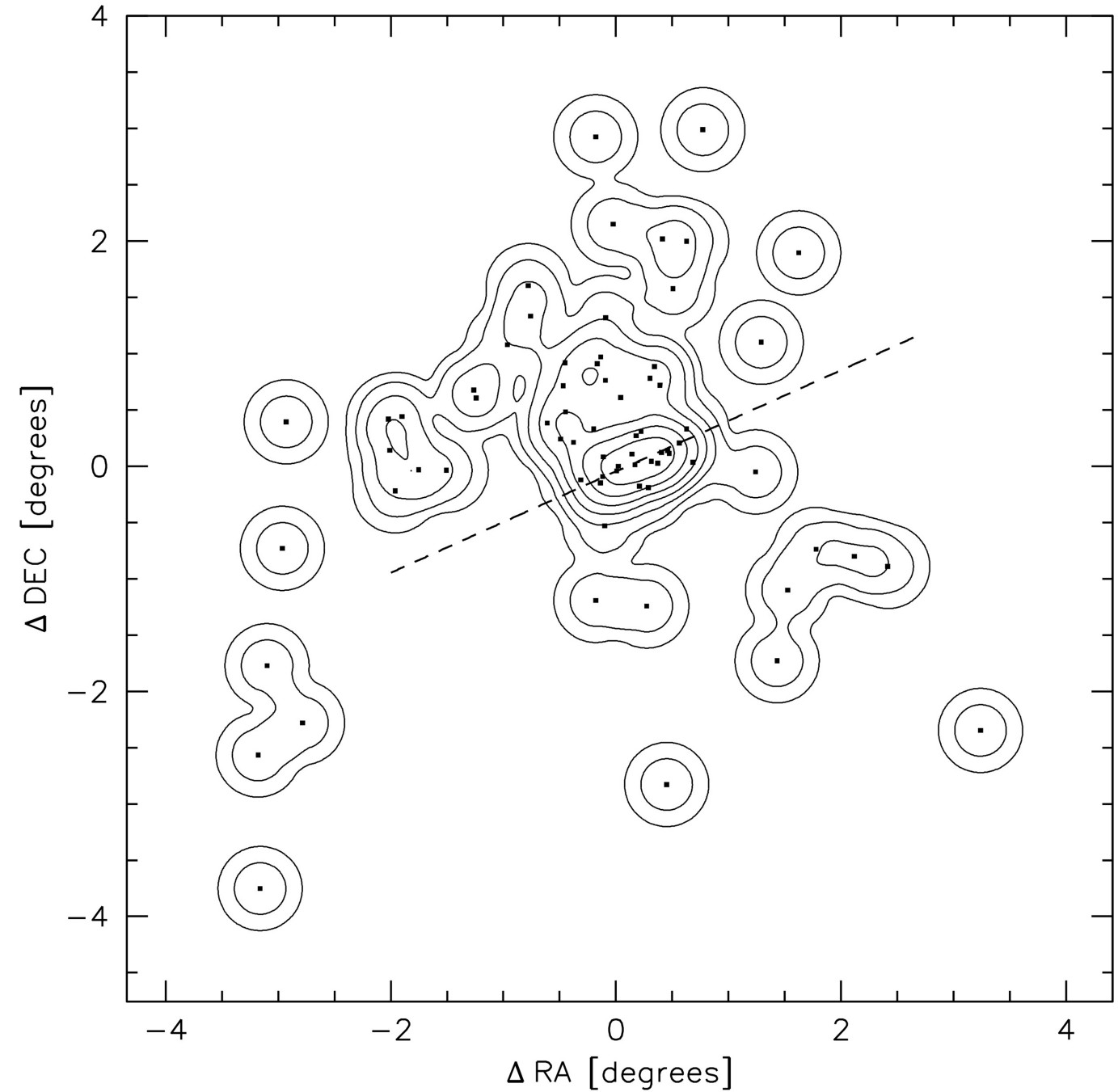

**Extended Data Fig. 2 | Morphology of the NGC 1052 group.** Distribution of 72 probable group members from a recent compilation of galaxies in the NGC 1052 field[23]. Contours were derived with the non-parametric kernel density method[37]. The dashed line indicates the trail. There is no evidence that the trail is in the general direction of large-scale structure in this field.

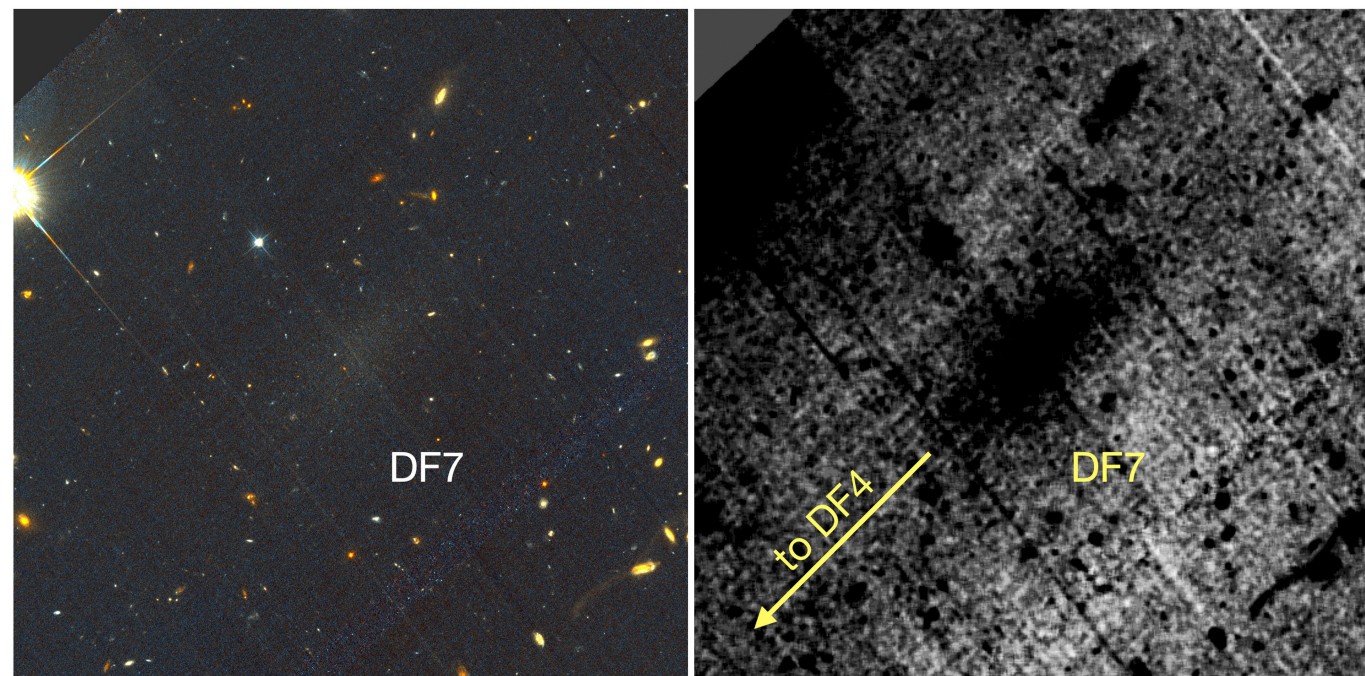

**Extended Data Fig. 3 | A candidate dark-matter-dominated galaxy.** HST/ACS images[1] of DF7, which is located at the western end of the galaxy trail, beyond DF4. Left: a colour image generated from the F606W and F814W data. Right: a median-filtered version of the F606W image, with the arrow depicting the direction towards DF4. DF7 is a candidate for the dark-matter-dominated remnant of one of the original galaxies[11], given its location and its elongation in the direction of DF4.

**Extended Data Table 1 | Coordinates of the candidate trail galaxies**

| Galaxy | RA [deg] | DEC [deg] |
|---|---|---|
| RCP32 | 40.6202 | -8.3768 |
| NGC1052-DF2 | 40.4451 | -8.4028 |
| RCP28 | 40.4215 | -8.3475 |
| RCP26 | 40.2897 | -8.2968 |
| RCP21 | 40.1200 | -8.2434 |
| NGC1052-DF9 | 40.0292 | -8.2290 |
| RCP17 | 39.9696 | -8.2121 |
| TA21-12000 | 39.9139 | -8.2285 |
| NGC1052-DF4 | 39.8128 | -8.1160 |
| NGC1052-DF5 | 39.8028 | -8.1408 |
| LEDA4014647 | 39.7021 | -8.0494 |
| NGC1052-DF7 | 39.6241 | -7.9257 |

Eleven galaxies were identified objectively with the Hough transform. NGC1052-DF9 (SDSS J024007.01–081344.4) is not in the sample of ref. [23] and was identified in a visual inspection.