## [Peer Review File · Nature]

Manuscript Title: A trail of dark matter-free galaxies from a bullet dwarf collision

Reviewer Comments & Author Rebuttals

Reviewer Reports on the Initial Version:

Referees' comments:

Referee #1 (Remarks to the Author):

The authors provide a tentative scenario to explain the existence and properties of the ultra diffuse galaxies DF2 and DF4, using a single (galaxy encounter) event some 6-9 Gyr ago. The authors base their claim on several pieces of observations, including: the apparent alignment of about 10 galaxies that include both DF2 and DF4, the relative large size of the galaxies identified as members of that 'linear structure', as well as other intriguing properties of those systems. The scenario proposed by the authors may be supported by simulation work (e.g., Shin et al. 2020) which predict the fragmentation of the galaxy progenitors, the violent redistribution of the respective 'components' (dark matter, versus gas, versus stars), and triggered star-formation.

The text and figures do a good job at illustrating the claim made by the authors. The scenario is interesting and, as mentioned, possibly supported by first principles and past hydro-dynamical simulations. The paper itself is mostly a gathering of already published information followed by interesting speculations, and the authors have done excellent work delivering a clear and well written flow.

The 'speculations' part is actually the main weakness of the paper: it contains a number of statements which are tentative ideas, and the authors cannot (yet) provide robust evidence for the proposed scenario (while they briefly provide some predictions; see comments below). There are indeed good hints that a trail of low surface brightness systems may be 'dynamically/historically' associated, but there is so far little in this paper to convince the reader that the scenario proposed by the authors 1- really works 2- is the most probable. It does not lower the value of such a paper (which may trigger further observations/simulations of the mentioned systems), but it does not qualify it, I believe, for a publication in Nature. Some more specific comments are provided below.

Remarks, questions:

- Generally speaking: the authors should ensure that their statements are not misleading, i.e. do depend on whether the proposed scenario is correct or not. For example, in the abstract, the authors write "we find that DF2 and DF4 are not alone but part of a linear substructure": this is correct per se, but possibly misleading if this means that it is a physical substructure. Rewording it to emphasise that this is an "apparent" substructure would, for example, clarify this (see other comments below).

- I may be missing something (apologies if I do) but I do not see how the authors can strictly say that the galaxies are on trajectories that are taking them away from each other along a vector that is tilted $7\text{deg} \pm 2$ from the line of sight. While the geometry depicted in Fig. 1 seems relevant, this does not guarantee that this statement is correct. This should be clarified.

- The sentence "Using their current (...) we find that they were together 5.7 Gyr ago, (...)" should be reworded. While the statement is correct when not including "possible velocity changes or motion in the plane of the sky", it is misleading at best. If the authors wish to convince the reader, they need to use sharper statements (in my opinion).

- "It has also been shown that the large present-day sizes of the galaxies": which "galaxies" are the authors talking about? Please clarify: is this a general statement, or something that relates to specific simulations (referred to in the text)?

- The sentence "In an encounter of two gas rich galaxies the collisional gas is shocked and separates (...)" should be rephrased. This is by far not strictly true and depends on many parameters.

- A personal comment that the authors should feel free to ignore: the authors write "We examine the spatial distribution (...) to further investigate this model.". It seems to me highly probable that the authors did identify the 'trail' of galaxies before investigating further. Hence, the logical flow of the paper reinforced by such a sentence seems, to me, quite misleading. Whether my interpretation is correct or not is not too important, and this, of course, does not remove the value of assessing, via some objective process (e.g. Hough transform), the existence of such an apparent trail more globally. I would just take this opportunity to kindly suggest to re-phrase this sentence if relevant, and look for similar misleading threads (if any). Again, just a personal request that the authors should feel free to ignore.

- While it is true that the los velocity difference between DF2 and DF4 is much larger than the velocity dispersion of the NGC1052 group, DF4's velocity is only ~ 40 km/s away from the systemic velocity of NGC1052. Hence, DF2 could be the red herring in this scenario. This should be clarified (here and all along the paper, as it seems like an important point).

- "This calculation assumes that galaxies are oriented randomly with respect to NGC1052". This is clearly not a good approximation in almost all sensible scenarios for the formation of low surface brightness galaxies. Then the joint probability (for the isotropic case) derived later on in that same paragraph is mostly irrelevant and misleading. This should be clarified.

- The statement 'it is not a coincidence that there are two galaxies': the authors go from 'are very likely connected' to 'it is not'. This should be fixed (it could be a coincidence, at the end).

- Fig 5:

* The authors do not comment much on the fact that the largest galaxies (right panel of Fig 5) are actually mostly near the three largest systems of the group. This was already emphasised by Roman et al. (their Figs. 11+12) without introducing the relative brightness of those systems. Considering that the picture may be biased by, for example, the proximity to NGC1052, it is not clear whether or

not the probability to obtain such a distribution is significant or not (while it is intriguing, at least from a first look).

* The fit to the running median is not representative of the brightest magnitude bin systems (right panel of Fig 5). While the systems within the 'trail' are mostly in the upper envelope of the distributions, such a fit over-emphasises the fact that only 1 out of 10 (filled) points in the left panel of Fig 5 is below the least squares fit.

[in other words: running a different fit, could exclude more than 1, possibly up to 3 galaxies, and make the right panel plot a bit less compelling]. I would suggest to check whether there is any bias in terms of the systems in the $20 > g > 19$ magnitude bin having more galaxies in the outskirts of the group (at larger distances from NGC1052) and clarify the interpretation of such plots accordingly. I would also suggest that the authors acknowledge the results from Roman et al. on the sizes of the low surface brightness galaxies within this field.

* Minor issue: I would advise the authors to use different symbols for the two (different) grouping in the left and right panels, as using open/filled circles in both may lead to confusion. It would actually be potentially a nice idea (and revealing?) to code things in such a way that you keep the magnitude and/or radii coding also in the right panel (using e.g. symbols sizes or colours).

- The authors briefly mention some predictions for the properties of the products from the suggested collision. This is quite useful and welcome in the context of such a speculative model. However, the level of details in such predictions is relatively low at present. Could the authors say something more specific about the velocities of each system in the trail (what is to be expected)? Could the authors propose predictions for the progenitors' masses or relative velocities at the time of the encounter? (and if not, what would be needed to constrain those). More generally speaking, I would suggest that the authors also make it very explicit what could validate their scenarios and what could confirm it, hence be as quantitative and specific as possible.

- The authors do not mention much regarding alternatives (scenarios) that may be viable (or not).

- The authors do not mention anything about the relative importance of the two more massive members of the group, including NGC1042. Rewinding back for about 6 to 9 Gyr: could this be an issue for the proposed scenario? (or not)

- The authors write "(...) this, in turn, demonstrates that dark matter is a substance, consisting of particles or objects (...)". This is a rather blunt statement, which should be removed. If this were so, then it would be an important milestone in demonstrating that all other alternatives are dismissed. As far as I can personally tell, it is by far not the case. (the same applies to the sentence about the "separable" quality, which itself relies on the fact that it can indeed be "separated").

Referee #2 (Remarks to the Author):

The paper by van Dokkum et al. titled "A trail of galaxies with and without dark matter from a single bullet dwarf collision" suggests that the ultra-diffuse galaxies DF2 and DF4 could have formed in a single collision event of gas-rich galaxies.

The study suggests is very timely and of high relevance for the broad community interested in LSBG or galaxy formation in general. As the authors mentioned in the manuscript, the complete information of the 11 objects on the proposed collision trail (3D position, cz, the velocity of each galaxy, and the dynamical mass) is indispensable to advocate this scenario as the origin of DF2 and DF4. I recommend this paper be published after the authors address the following issues.

1)

Hough's transformation suggested that the 11 objects lie in a straight line. To take the result as evidence for the scenario, the catalog used in Hough transformation should be sampled without bias. However, selecting lower surface brightness objects includes a visual inspection, which is not parametrizable. It would be helpful if the authors explicitly present how the catalogs from Roman et al. 2021 are chosen.

2)

The authors tentatively suggested RCP32 and DF7 at the end of the trail as the precursors of the "bullet dwarf" collision. In the bullet dwarf collision scenario, however, most gas would have been left at the collision site while forming shock-induced stars. If so, the brightest and heaviest dark matter deficient objects would have been created in the middle part of the trail, not where DF2 and DF4 are (near the end of the trail, close to the precursor galaxies). I would like to hear the authors' thoughts on this.

3)

In addition, the precursor galaxies in the scenario are assumed to be gas-rich dwarfs before the collision, implying that considerable stars would have formed before the collision. However, both RCP32 and DF4 are fainter than DF2 and DF4. Meanwhile, according to Shin et al. (2020), galaxy collision simulations with velocities up to <700 km/s could form DMDGs in their collisional parameter study. There is a chance that the precursor galaxies might have collided with a velocity much higher than 348 km/s and have already escaped the 2 Mpc trail.

For these reasons, I wonder if the authors are willing to consider the possibility that the precursor galaxies may not be among the 11 objects. I also am curious to know if the authors plan to search for the precursor candidates in an extended field-of-view away from the trail.

4)

The authors suggest that the extended system of 11 objects (including DF2 and DF4) has experienced two very unlikely incidents: (1) high-speed head-on galaxy collision, and (2) the collision remnant (i.e., the trail of 11 objects) not having been affected by any other massive objects for 6-9 Gyr after the collision. It would be helpful if the authors could comment on the possibility of this coincidence.

Author Rebuttals to Initial Comments:

Comments from referee 1:

The authors provide a tentative scenario to explain the existence and properties of the ultra diffuse galaxies DF2 and DF4, using a single (galaxy encounter) event some 6-9 Gyr ago. The authors base their claim on several pieces of observations, including: the apparent alignment of about 10 galaxies that include both DF2 and DF4, the relative large size of the galaxies identified as members of that 'linear structure', as well as other intriguing properties of those systems. The scenario proposed by the authors may be supported by simulation work (e.g., Shin et al. 2020) which predict the fragmentation of the galaxy progenitors, the violent redistribution of the respective 'components' (dark matter, versus gas, versus stars), and triggered star-formation.

The text and figures do a good job at illustrating the claim made by the authors. The scenario is interesting and, as mentioned, possibly supported by first principles and past hydro-dynamical simulations. The paper itself is mostly a gathering of already published information followed by interesting speculations, and the authors have done excellent work delivering a clear and well written flow.

The 'speculations' part is actually the main weakness of the paper: it contains a number of statements which are tentative ideas, and the authors cannot (yet) provide robust evidence for the proposed scenario (while they briefly provide some predictions; see comments below). There are indeed good hints that a trail of low surface brightness systems may be 'dynamically/historically' associated, but there is so far little in this paper to convince the reader that the scenario proposed by the authors 1- really works 2- is the most probable. It does not lower the value of such a paper (which may trigger further observations/simulations of the mentioned systems), but it does not qualify it, I believe, for a publication in Nature. Some more specific comments are provided below.

We appreciate the concern of the referee. Our paper is certainly unusual, in that it is an observational paper that includes no new data! Nevertheless, two "observations" are new, apart from the interpretation.

The first is that DF2 and DF4 are moving away from each other with a large velocity along the line of sight. We, or others, could have realized this when we measured the TRGB distances to both galaxies (in Shen et al. 2021), but we missed this simple fact. Back then we did not understand why the galaxies were 2.1 Mpc apart from each other, five times the virial radius of NGC1052. In fact, we were suspicious of our result, and we waited with publication until we had done a lot of extra checks. A joint origin of the galaxies in a high speed encounter is not a conjecture, but a consequence of their present-day positions and velocities. Furthermore, in any other formation model there is no explanation for the large distance between the galaxies and their large velocity difference (both much larger than between any other galaxies in the group). We have made this clearer in the first part of the text. We now begin by stating the assumption that the two galaxies have a common origin, and show that the collisional origin then follows directly from the velocities and 3D positions of the galaxies. We also address the "does it really work" question in the revised version, by showing an example geometry of the orbits of the progenitor galaxies (as explained below).

The second is the striking alignment of galaxies along the DF2 - DF4 axis. Even though the catalog that we use was previously published (Roman et al 2021), the authors of that paper missed this alignment. Importantly, we did *not* find this alignment and then looked for a model to explain it, as the referee perhaps (logically) suspects. We first derived the joint formation of DF2 and DF4 from their current positions and velocities, and only then plotted positions of other galaxies to look for any other objects along this line. (as described in the paper - see below for our response to the referee's question about the chronological order of the discovery of the alignment)

Having said that, we agree that it is crucial for any explanation that it provides predictions that can be used to falsify the model. We have sharpened this part of the paper, providing specific predictions for the velocities and line-of-sight distances of the galaxies that are part of the trail. We also provide a prediction for the ages of the globular clusters in DF4, which have not yet been measured. They should be identical to those in DF2, that is, have the same unusual (young, for their metallicity) age. A direct observational test is to compare an averaged spectrum of DF4 globular clusters to the (published) averaged spectrum of DF2 globular clusters. Our model predicts that these spectra will turn out to be identical.

We hope that these responses, and the changes to the paper, address the concerns of the referee. Again, we fully appreciate that this is an unusual paper, and we understand the referee's hesitation.

Remarks, questions:

- Generally speaking: the authors should ensure that their statements are not misleading, i.e. do depend on whether the proposed scenario is correct or not. For example, in the abstract, the authors write "we find that DF2 and DF4 are not alone but part of a linear substructure": this is correct per se, but possibly misleading if this means that it is a physical substructure. Rewording it to emphasise that this is an "apparent" substructure would, for example, clarify this (see other comments below).

We added the word "apparent", as suggested by the referee.

- I may be missing something (apologies if I do) but I do not see how the authors can strictly say that the galaxies are on trajectories that are taking them away from each other along a vector that is tilted $7\text{deg}\pm 2$ from the line of sight. While the geometry depicted in Fig. 1 seems relevant, this does not guarantee that this statement is correct. This should be clarified.

The galaxies are certainly currently moving away from each other: the distance between them, as viewed from Earth, is increasing. However, we agree that we do not know the velocity in the plane of the sky, and the referee is correct in that this statement ultimately relies on the assumption that the galaxies have a common origin. As noted above we have rephrased the beginning of the paper, and we now begin by stating this assumption. We removed the "along a vector that is tilted 7 degree" text from the main text (we kept the 7 degrees in the abstract, as there it simply describes the current geometry) and generally toned down the statements in the opening paragraph. In the third paragraph we now present an example collision scenario involving NGC1052 (see below), and it is now only there that we estimate the time since the collision.

- The sentence "Using their current (...) we find that they were together 5.7 Gyr ago, (...)" should be reworded. While the statement is correct when not including "possible velocity changes or motion in the plane of the sky", it is misleading at best. If the authors wish to convince the reader, they need to use sharper statements (in my opinion).

We agree - see our response to the previous comment, and the response to the comment below about the velocities of DF2 and DF4 with respect to NGC1052.

- "It has also been shown that the large present-day sizes of the galaxies": which "galaxies" are the authors talking about? Please clarify: is this a general statement, or something that relates to specific simulations (referred to in the text)?

This was indeed confusing. The sentence now reads:

“Although galaxies that form this way are initially compact, supernova feedback is expected to increase their sizes, an effect that is particularly efficient when the stars are not bound by a dark matter halo (Trujillo-Gomez et al. 21ab).”

- The sentence "In an encounter of two gas rich galaxies the collisional gas is shocked and separates (...)" should be rephrased. This is by far not strictly true and depends on many parameters.

Yes - we were describing a particular model, but that was not clear. We rephrased it as "In a near head-on collision between two gas rich galaxies the collisional gas can be shocked and separated from the collisionless dark matter and pre-existing stars (Silk 2019; Shin et al. 2020).”

- A personal comment that the authors should feel free to ignore: the authors write "We examine the spatial distribution (...) to further investigate this model.". It seems to me highly probable that the authors did identify the 'trail' of galaxies before investigating further. Hence, the logical flow of the paper reinforced by such a sentence seems, to me, quite misleading. Whether my interpretation is correct or not is not too important, and this, of course, does not remove the value of assessing, via some objective process (e.g. Hough transform), the existence of such an apparent trail more globally. I would just take this opportunity to kindly suggest to re-phrase this sentence if relevant, and look for similar misleading threads (if any). Again, just a personal request that the authors should feel free to ignore.

We understand why the referee is suspicious, but it is true: we first came to the realization that DF2 and DF4 are speeding away from each other and could have a common, collisional origin, then realized the striking similarity to the initial conditions in Silk (2019) and subsequent simulation papers, and only then investigated whether there were other galaxies along the DF2-DF4 axis. This all happened over the course of a few days, and the lead author kept the others informed of these developments as they happened. It really was an “aha moment” when we plotted the spatial distribution of the Roman et al. sample and the line of galaxies jumped out at us! This truly was a prediction, or expectation, that was then immediately confirmed thanks to the excellent catalog that Roman and collaborators put together.

We would like to retain this chronological, narrative aspect in the paper, as it adds weight to the interpretation and it is what actually happened.

- While it is true that the los velocity difference between DF2 and DF4 is much larger than the velocity dispersion of the NGC1052 group, DF4's velocity is only ~40 km/s away from the systemic velocity of NGC1052. Hence, DF2 could be the red herring in this scenario. This should be clarified (here and all along the paper, as it seems like an important point).

We agree, and we thank the referee for pointing out that DF2 likely was the main culprit, in the sense that it had the higher initial velocity with respect to the group. We examined possible geometries that would satisfy the velocity constraints, also inspired by Shin et al. (2020) (in particular their Fig 7). We came up with an example scenario that fits all available constraints. Progenitor 1 fell into the group with a high velocity and progenitor 2 was bound to NGC1052. After the collision one remnant continued on its unbound orbit whereas the other fell back toward NGC1052. The dark matter free fragments are distributed in between the two remnants, with DF2 close to the remnant of progenitor 1 and DF4 close to progenitor 2.

This is now shown in Fig 1b. This is just an illustration; a full exploration of parameter space in a simulation is beyond the scope of the paper, but this is at least an improvement over the simplistic straight line that we had before. It is also quite reassuring how this model explains the relative velocities of NGC1052, DF2, and DF4; provides an even better match to the spectroscopic age of DF2 (9 Gyr); and is similar to the examples shown in Shin et al. (2020).

Again we thank the referee for this comment: the asymmetry in the velocities (with DF4 close to NGC1052 but DF2 very discrepant) was the main weakness in our model, and it looks like there is a plausible explanation for it that makes the paper significantly better.

- "This calculation assumes that galaxies are oriented randomly with respect to NGC1052". This is clearly not a good approximation in almost all sensible scenarios for the formation of low surface brightness galaxies. Then the joint probability (for the isotropic case) derived later on in that same paragraph is mostly irrelevant and misleading. This should be clarified.

It is true that the spatial distribution of satellite galaxies generally shows some flattening, but it is not a strong effect: the average projected axis ratio of SDSS groups is 0.77 (Wang et al. 2008). Furthermore, the NGC1052 group has a global orientation that is, if anything, perpendicular to the trail. This is shown in a new figure in the Methods section, included here for convenience:

The points and contours show the distribution of all 72 group galaxies (the faint galaxies plus the bright galaxies with $cz < 2000$ km/s) from Roman et al. Apart from the 11 galaxies of the trail in the center of the group there are no galaxies along the same axis at larger radii, as would have been expected if this were the general direction of the large scale structure in this field. Assuming isotropy for the randomized samples is, if anything, somewhat conservative, as it will scatter more galaxies toward the line than away from the line.

We explain this now in a new Methods section.

- The statement 'it is not a coincidence that there are two galaxies': the authors go from 'are very likely connected' to 'it is not!'. This should be fixed (it could be a coincidence, at the end).

We agree, and we also see that the point of this statement was not very clear. We changed it to: "Before turning to the properties of the galaxies in the trail we note that the Hough transform provides post-hoc validation of our initial assumption that DF2 and DF4 are related to each other." (the paper now starts with this explicit assumption)

- Fig 5:

* The authors do not comment much on the fact that the largest galaxies (right panel of Fig 5) are actually mostly near the three largest systems of the group. This was already emphasised by Roman et

al. (their Figs. 11+12) without introducing the relative brightness of those systems. Considering that the picture may be biased by, for example, the proximity to NGC1052, it is not clear whether or not the probability to obtain such a distribution is significant or not (while it is intriguing, at least from a first look).

The brightest galaxies are near the center of the group, as is the trail, and it is not clear whether there is a significance beyond that. We now note that Roman et al. (and also Cohen et al. 2018) already found that the largest galaxies are near the center of the group:

“The unusual prevalence of large, low surface brightness galaxies in the central regions of the NGC1052 group had been noticed previously (Cohen et al. 2018; Roman et al. 2021); here we propose that the bullet dwarf event was responsible for it.”

* The fit to the running median is not representative of the brightest magnitude bin systems (right panel of Fig 5). While the systems within the ‘trail’ are mostly in the upper envelope of the distributions, such a fit over-emphasises the fact that only 1 out of 10 (filled) points in the left panel of Fig 5 is below the least squares fit.

[in other words: running a different fit, could exclude more than 1, possibly up to 3 galaxies, and make the right panel plot a bit less compelling]. I would suggest to check whether there is any bias in terms of the systems in the $20 > g > 19$ magnitude bin having more galaxies in the outskirts of the group (at larger distances from NGC1052) and clarify the interpretation of such plots accordingly. I would also suggest that the authors acknowledge the results from Roman et al. on the sizes of the low surface brightness galaxies within this field.

This is a good point. We addressed it by following the referee’s suggestion to their next point, namely to plot a continuous scale in panel b rather than a binary “above or below the fit” (which is indeed sensitive to the precise fit).

* Minor issue: I would advise the authors to use different symbols for the two (different) grouping in the left and right panels, as using open/filled circles in both may lead to confusion. It would actually be potentially a nice idea (and revealing?) to code things in such a way that you keep the magnitude and/or radii coding also in the right panel (using e.g. symbols sizes or colours).

We agree and thank the referee for the suggestion. We changed the right panel, color coding the points according to their radius (with respect to the linear fit in the left panel). This makes the same point as before, preserving more information and without a somewhat arbitrary cut.

- The authors briefly mention some predictions for the properties of the products from the suggested collision. This is quite useful and welcome in the context of such a speculative model. However, the level of details in such predictions is relatively low at present. Could the authors say something more specific about the velocities of each system in the trail (what is to be expected)? Could the authors propose predictions for the progenitors' masses or relative velocities at the time of the encounter? (and if not, what would be needed to constrain those). More generally speaking, I would suggest that the authors also make it very explicit what could validate their scenarios and what could confirm it, hence be as quantitative and specific as possible.

We rewrote this paragraph, now providing equations relating the positions of galaxies on the trail to their predicted radial velocities and TRGB distances. We also added a prediction that could falsify the joint formation of DF2 and DF4 in a model-independent way, namely a comparison between the averaged spectra of globular clusters in DF2 and DF4. The DF2 clusters have been observed but the DF4 clusters have not (the only spectra that have been published are high resolution and in the red, for radial velocities, and these don’t give information on the stellar populations). The DF4 spectra are predicted to

be identical to those of DF2, something that only makes sense for these galaxies (that are 2 Mpc apart!) if they formed together.

- The authors do not mention much regarding alternatives (scenarios) that may be viable (or not).

We now added a subsection to the Methods section where we list some of the scenarios that have been proposed. Most center on the absence of dark matter only, and invoke some form of extreme tidal interaction (with NGC1052 or other galaxies) to strip the dark matter away. All these models have the same weaknesses: they do not explain (or even attempt to explain) the unusual, overluminous globular clusters, and they do not explain why there are two near-identical galaxies in the same group. Besides the bullet scenario, the only model that explains the clusters is that of Trujillo-Gomez et al, who begin their analysis with scatter in the halo mass - stellar mass relation and study how star formation proceeds in the most dark matter-deficient objects. Their model has ad hoc initial conditions and does not account for the presence of two galaxies, but the key aspects of it (the formation of globular clusters, and the subsequent puffing up of the galaxies due to feedback) likely apply to the collision products in the bullet scenario.

- The authors do not mention anything about the relative importance of the two more massive members of the group, including NGC1042. Rewinding back for about 6 to 9 Gyr: could this be an issue for the proposed scenario? (or not)

We are not sure what to say about these galaxies. NGC1052 contains the vast majority of the dark and luminous matter in the group, and it is unlikely that NGC1047, NGC1035, or NGC1042 influenced the dynamics of the collision unless they happened to be intersecting the orbits.

- The authors write "(...) this, in turn, demonstrates that dark matter is a substance, consisting of particles or objects (...)". This is a rather blunt statement, which should be removed. If this were so, then it would be an important milestone in demonstrating that all other alternatives are dismissed. As far as I can personally tell, it is by far not the case. (the same applies to the sentence about the "separable" quality, which itself relies on the fact that it can indeed be "separated").

We agree that these sentences were a bit grandiose and removed them. We now only compare to the bullet cluster, and note the importance of finding examples on small scales to further constrain the dark matter cross section.

Referee #2 (Remarks to the Author):

The paper by van Dokkum et al. titled "A trail of galaxies with and without dark matter from a single bullet dwarf collision" suggests that the ultra-diffuse galaxies DF2 and DF4 could have formed in a single collision event of gas-rich galaxies.

The study suggests is very timely and of high relevance for the broad community interested in LSBG or galaxy formation in general. As the authors mentioned in the manuscript, the complete information of the 11 objects on the proposed collision trail (3D position, cz, the velocity of each galaxy, and the dynamical mass) is indispensable to advocate this scenario as the origin of DF2 and DF4. I recommend this paper be published after the authors address the following issues.

We thank the referee for their kind words! We note that we made several additions to the paper in response to comments from referee 1. We hope that both referees like the changes.

1)

Hough's transformation suggested that the 11 objects lie in a straight line. To take the result as evidence for the scenario, the catalog used in Hough transformation should be sampled without bias. However, selecting lower surface brightness objects includes a visual inspection, which is not parametrizable. It would be helpful if the authors explicitly present how the catalogs from Roman et al. 2021 are chosen.

We thank the referee for the suggestion, and have added this information to the Methods section. While it is true that visual inspections cannot easily be replicated, the Roman et al catalog is publicly available and so the analysis in our paper is at least repeatable. On a subjective note, we visually inspected our own deep Dragonfly images of the same area, and confirm all these objects (and do not find any that were missed).

2)

The authors tentatively suggested RCP32 and DF7 at the end of the trail as the precursors of the "bullet dwarf" collision. In the bullet dwarf collision scenario, however, most gas would have been left at the collision site while forming shock-induced stars. If so, the brightest and heaviest dark matter deficient objects would have been created in the middle part of the trail, not where DF2 and DF4 are (near the end of the trail, close to the precursor galaxies). I would like to hear the authors' thoughts on this.

The distribution of systems along the trail depends on the detailed geometry of the encounter and the structure of the progenitors. For instance, it will matter greatly if this was a grazing or truly head-on encounter. In the former, the massive structures would be situated near the end points, whereas in the latter they would indeed be near the center. We note that in one of the simulations of Shin et al. (2020) the most massive clump is *not* in the center, but a lower mass one is (Fig. 7 of that paper). We contacted Eun-jin Shin and asked about this, and he saw no particular objections to having the most massive clumps be the furthest removed. Turning this around, the fact that there are two near-identical massive clumps near the ends of the trail can probably be used to further constrain the geometry of the impact event.

3)

In addition, the precursor galaxies in the scenario are assumed to be gas-rich dwarfs before the collision, implying that considerable stars would have formed before the collision. However, both RCP32 and DF4 are fainter than DF2 and DF4. Meanwhile, according to Shin et al. (2020), galaxy collision simulations with velocities up to <700 km/s could form DMDGs in their collisional parameter study. There is a chance that the precursor galaxies might have collided with a velocity much higher than 348 km/s and have already escaped the 2 Mpc trail.

For these reasons, I wonder if the authors are willing to consider the possibility that the precursor galaxies may not be among the 11 objects. I also am curious to know if the authors plan to search for the precursor candidates in an extended field-of-view away from the trail.

First, we note that we now included a plausible scenario where one progenitor was on an unbound orbit and the other was satellite of NGC1052. This model explains the relative velocities of DF2 and DF4 with respect to NGC1052, and places the analysis on a firmer footing.

In answer to the referee's question, we carefully inspected our Dragonfly images as well as the DECaLS data beyond the trail and did not find any other candidates for the progenitors. This can also be seen in the new Extended Data Figure 1, which shows the large scale distribution of galaxies in this area.

In terms of the brightness of the remnants, Silk (2019) expected them to be nearly completely dark, as they would not have formed many stars prior to the collision. It is interesting that RCP32 is much fainter than DF7: RCP32 is the plausible remnant of the infalling object, and it makes sense that it is fainter than the originally-bound galaxy DF7.

Having said all that, the identification of RCP32/DF7 with the remnants is probably the most speculative part of our study, and we have been careful to say "tentatively identified" in the abstract and the main text.

4)

The authors suggest that the extended system of 11 objects (including DF2 and DF4) has experienced two very unlikely incidents: (1) high-speed head-on galaxy collision, and (2) the collision remnant (i.e., the trail of 11 objects) not having been affected by any other massive objects for 6-9 Gyr after the collision. It would be helpful if the authors could comment on the possibility of this coincidence.

We agree, and (partly inspired by Shin et al. 2020) we now propose a scenario where only one of the progenitors fell into the group on an unbound orbit. These events are rare but not exceptionally so: based on Shin et al.'s analysis we estimate that there are ~8 within a sphere of 20 Mpc. Furthermore, we take the effects of NGC1052 now into account (see the new Fig 1b); the trail will remain intact as long as the galaxies nearest to NGC1052 did not complete more than ~half an orbit.

Reviewer Reports on the First Revision:

Referees' comments:

Referee #1 (Remarks to the Author):

The authors have revised the manuscript taking into account most of the concerns and comments I made in my first report. (I feel that -) The updated text better emphasises the assumptions made by the authors, while clearly conveying the speculative nature of their claims. It also focuses on the puzzling observables more directly leading to the proposed formation scenario.

I believe the paper is a useful and stimulating addition to the literature on this particular system. It should trigger new follow-up studies to confirm or infirm this scenario. If correct, it would be quite a story to tell, and would potentially shine a new light on other systems. If wrong, it may further motivate new efforts to understand the NGC1052 group, and possibly trigger a search for such collisional events. I therefore believe it deserves to be published.

I still see the paper as highly speculative (which does ring as a positive bell in my mind), given the still narrow body of evidence. If the paper had included a second confirmation level, it would naturally fit within the realm of papers published in Nature. I personally feel that it still lacks such a confrontation with facts. However, I believe it is up to the editor to decide whether such a format is fit for a publication in Nature or not.

Either way, I include two comments that echo my original report, and that I feel the authors should seriously consider, before publishing (in Nature or elsewhere).

1- Time travel

On my comment about 'rewinding time' taking into account the other members of this group, the authors state that they are "not sure what to say" and that it is unlikely impacting the proposed scenario "unless they happened to be intersecting the orbits".

This is a bit of an odd statement considering a subset of the authors did publish a research note (Research Notes of the American Astronomical Society, Volume 3, 2, 29) on the distance to NGC1042, where they wrote "a likely interpretation of Fig. 1 is that NGC 1052 stripped gas from NGC 1042 in a past interaction."

This should be clarified.

2- Credit

While Shin et al. did not single out the "linear structure" mentioned by the authors, they did explicitly mention the potential of such a collisional scenario to form DF2 and DF4. I believe it would be more than fair to clearly and positively acknowledge it in the manuscript, beyond the present citations (which refer to other matters, such as shocks, low probability of such collisions, geometries, formation of more than two galaxies). [Let me emphasise that I am not a co-author of the Shin et al. paper, and declare no conflicting - or any existing - ties with that group or publication]

Referee #2 (Remarks to the Author):

I appreciate the changes and that the authors added specific predictions and comments for the velocity and the distance of objects in the trail.

I would be ready to recommend the manuscript for publication after seeing the responses by the authors on the following (minor) issues:

1) The authors strongly assert that the bullet dwarf event is a persuasive scenario for the formation of NGC1052-DF2 and DF4 — the only scenario that explains the multiple dark matter-free objects with luminous GCs. Readers might wonder if there is any counter-evidence to this scenario in the observation. For example, the simulations in Lee et al. (2021) do not show the diffuse nature of dm-free objects. Their results also apparently conflict with the sentence, "Galaxies that form this way are initially compact, but supernova feedback is expected to increase their sizes." I would like to hear how the authors think about this.

2) The authors wrote, "In this model, DF2 has traveled ~ 3 Mpc since the collision with a velocity of 315km/s". The value 315km/s is the observed relative velocity to the host in the present day; therefore, the mention is incorrect.

In the model shown in Fig 1b, DF2 and DF4 are affected by different gravitational forces due to the different distances from the host galaxy, which also can affect the relative velocity of DF2 and DF4. In the model shown in Fig1b, the gravitational interaction by the host also might have been increased the relative velocity between DF2 and DF4.

3) The authors mentioned that DF1 could be a part of the trail formed by the same event. It will be helpful for readers to recognize the location of DF1 if the authors present the location of DF1 in Fig 3c or Fig 4 with a notation.

Author Rebuttals to First Revision:

We would like to thank the referees for their new reports. Detailed responses to the points that they brought up are given below.

Comments are in black; our responses are in blue.

Comments from referee 1:

The authors have revised the manuscript taking into account most of the concerns and comments I made in my first report. (I feel that -) The updated text better emphasises the assumptions made by the authors, while clearly conveying the speculative nature of their claims. It also focuses on the puzzling observables more directly leading to the proposed formation scenario.

I believe the paper is a useful and stimulating addition to the literature on this particular system. It should trigger new follow-up studies to confirm or infirm this scenario. If correct, it would be quite a story to tell, and would potentially shine a new light on other systems. If wrong, it may further motivate new efforts to understand the NGC1052 group, and possibly trigger a search for such collisional events. I therefore believe it deserves to be published.

I still see the paper as highly speculative (which does ring as a positive bell in my mind), given the still narrow body of evidence. If the paper had included a second confirmation level, it would naturally fit within the realm of papers published in Nature. I personally feel that it still lacks such a confrontation with facts. However, I believe it is up to the editor to decide whether such a format is fit for a publication in Nature or not.

Either way, I include two comments that echo my original report, and that I feel the authors should seriously consider, before publishing (in Nature or elsewhere).

1- Time travel

On my comment about 'rewinding time' taking into account the other members of this group, the authors state that they are "not sure what to say" and that it is unlikely impacting the proposed scenario "unless they happened to be intersecting the orbits".

This is a bit of an odd statement considering a subset of the authors did publish a research note (Research Notes of the American Astronomical Society, Volume 3, 2, 29) on the distance to NGC1042, where they wrote "a likely interpretation of Fig. 1 is that NGC 1052 stripped gas from NGC 1042 in a past interaction."

This should be clarified.

Apologies for being unclear. It is likely that some of the massive members of the group interacted with one another in the past. In particular, as the referee notes, the spiral galaxy NGC1042 likely had an encounter with the massive elliptical galaxy NGC1052. Whether this interaction happened at the same time as the formation of the trail is impossible to say; a priori it seems unlikely as the time scales for these interactions are much shorter than the age of the universe.

Nevertheless, we now appreciate the excellent point of the referee, namely that it is possible that another group member were involved or instrumental in setting up the initial conditions; e.g., the two progenitors could have been bound to different galaxies (NGC1052 and NGC1042) rather than bound to NGC1052 and unbound. Rather than saying "unbound" we now include this possibility explicitly, with a reference to the Research Note on NGC1042:

“We assume that the second progenitor was not bound to NGC1052, and either on first infall or a satellite of another massive galaxy in the group (van Dokkum et al. 2019).”

2- Credit

While Shin et al. did not single out the "linear structure" mentioned by the authors, they did explicitly mention the potential of such a collisional scenario to form DF2 and DF4. I believe it would be more than fair to clearly and positively acknowledge it in the manuscript, beyond the present citations (which refer to other matters, such as shocks, low probability of such collisions, geometries, formation of more than two galaxies). [Let me emphasise that I am not a co-author of the Shin et al. paper, and declare no conflicting - or any existing - ties with that group or publication]

We appreciate the point - from the abstract one might think that Shin et al. merely suggested that dark matter free galaxies could form this way, and not specifically DF2 and DF4. We tried to highlight the following distinction: although Shin et al. note that multiple clumps could form in the collision besides the main clump they do not suggest specifically that both DF2 and DF4 were formed at the same time, in the aftermath of a single collision. One of us (PvD) discussed this point with one of the authors of the paper (Joohyun Lee, who is the lead author on their follow-up higher resolution paper Lee et al. 2021), and he confirmed that this joint formation had not occurred to them (but also that it is very plausible).

In any case, one could argue that this is splitting hairs, and we certainly should not be credited with the general scenario of forming DF2 and/or DF4 in a collision of this kind. We therefore changed the abstract to make this explicit, and also now state that Shin et al. already saw multiple clumps form in their simulation:

“It has been suggested that these galaxies [changed from “such galaxies”, now referring explicitly to DF2 and DF4] were formed in the aftermath of high velocity collisions of gas rich galaxies, \cite{silk:19,shin:20,lee:21} events that resemble the collision that created the bullet cluster\cite{clowe:06} but on much smaller scales. The gas separates from the dark matter in the collision and subsequent star formation leads to the formation of one or more dark matter-free galaxies. \cite{shin:20}” [added this last sentence]

and then the following sentence:

“Here we show that the present-day line-of-sight distances and radial velocities of DF2 and DF4 are consistent with their joint formation in the aftermath of a single collision of this kind, $\sim 9\text{ Gyr}$ ago. Moreover, ...”

and then we describe the trail.

We made a similar change to the text. The phrase “.. scenarios that have previously been proposed for the formation of individual dark matter-less, globular cluster-rich galaxies such as DF2 and DF4.” is now replaced by the more direct

“.. scenarios that have previously been proposed for the formation of DF2 and DF4.”

Referee #2 (Remarks to the Author):

I appreciate the changes and that the authors added specific predictions and comments for the velocity and the distance of objects in the trail.

I would be ready to recommend the manuscript for publication after seeing the responses by the authors on the following (minor) issues:

1) The authors strongly assert that the bullet dwarf event is a persuasive scenario for the formation of NGC1052-DF2 and DF4 — the only scenario that explains the multiple dark matter-free objects with luminous GCs. Readers might wonder if there is any counter-evidence to this scenario in the observation. For example, the simulations in Lee et al. (2021) do not show the diffuse nature of dm-free objects. Their results also apparently conflict with the sentence, "Galaxies that form this way are initially compact, but supernova feedback is expected to increase their sizes." I would like to hear how the authors think about this.

We agree that we combined two things here without explaining them very well: the simulations in Lee et al. (2021), which show that compact dark matter-free galaxies form with an excess of globular clusters, and independent work by Trujillo-Gomez et al. (2021) who showed that feedback from globular cluster formation in dark matter-deficient galaxies leads to an increase of the galaxies' sizes. We made this clearer in the text, which now reads:

"Galaxies that form this way are initially compact, \cite{lee:21} in apparent conflict with the large half light-radii of DF2 and DF4. However, intense feedback accompanying the formation of the globular clusters is expected to increase the sizes of the newly formed galaxies, an effect that is particularly efficient when the stars are not bound by a dark matter halo. \cite{trujillogomez:21,trujillogomez:21b}"

We also made the illustration of the proposed scenario (formerly figure 2, now Extended Data Figure 1) clearer, showing better that the galaxies probably were initially compact and then expanded.

2) The authors wrote, "In this model, DF2 has traveled ~ 3 Mpc since the collision with a velocity of 315km/s". The value 315km/s is the observed relative velocity to the host in the present day; therefore, the mention is incorrect.

In the model shown in Fig 1b, DF2 and DF4 are affected by different gravitational forces due to the different distances from the host galaxy, which also can affect the relative velocity of DF2 and DF4. In the model shown in Fig1b, the gravitational interaction by the host also might have been increased the relative velocity between DF2 and DF4.

We thank the referee for noting this, as it is indeed probable that there have been changes to the velocities of both DF2 and DF4 with respect to NGC1052. In the model of Fig 1b, the change to DF4 is quite significant, due to its orbit: it had a positive velocity with respect to NGC1052 at the time of the collision whereas it now has a negative velocity. The change to DF2's velocity since the collision was much smaller, as it has never been bound to NGC1052 in this scenario, but it is indeed likely that its velocity was higher immediately after the collision. We changed the text to the following:

"In this model DF2 has traveled ~3 Mpc since the collision. In this model DF2 has traveled ~3 Mpc since the collision with an average velocity with respect to NGC1052 that is likely somewhat higher than its present-day value of 315 kms. Assuming $\langle v \rangle \sim 350$ km/s dates the collision to ~8 Gyr ago, in excellent agreement with the ages of the globular clusters and the diffuse light in DF2 (9+-2 Gyr, as measured from optical spectra \cite{dokkum:18b,fensch:19})."

We changed the age of the collision from ~9 Gyr to ~8 Gyr in the abstract and Fig 1 accordingly.

3) The authors mentioned that DF1 could be a part of the trail formed by the same event. It will be helpful for readers to recognize the location of DF1 if the authors present the location of DF1 in Fig 3c or Fig 4 with a notation.

This is a good point; we added the label to Fig. 4b (the old Fig 5), where it is in context with Fig. 4a. For convenience, and to avoid confusion, we also added a Table to the Methods section with the coordinates of all trail galaxies.